# Nopol-Based Quinoline Derivatives as Antiplasmodial Agents

**DOI:** 10.3390/molecules26041008

**Published:** 2021-02-14

**Authors:** Rogers J. Nyamwihura, Huaisheng Zhang, Jasmine T. Collins, Olamide Crown, Ifedayo Victor Ogungbe

**Affiliations:** Department of Chemistry, Physics, and Atmospheric Sciences, Jackson State University, Jackson, MS 39217, USA; rogersnyamwihura@yahoo.com (R.J.N.); huaishengzhang@medicilon.com.cn (H.Z.); jasmine.collins@wvstateu.edu (J.T.C.); olamide.crown@jsums.edu (O.C.)

**Keywords:** *Plasmodium falciparum*, aminoquinoline, nopol, *α*-pinene, malaria

## Abstract

Malaria remains a significant cause of morbidity and mortality in Sub-Saharan Africa and South Asia. While clinical antimalarials are efficacious when administered according to local guidelines, resistance to every class of antimalarials is a persistent problem. There is a constant need for new antimalarial therapeutics that complement parasite control strategies to combat malaria, especially in the tropics. In this work, nopol-based quinoline derivatives were investigated for their inhibitory activity against *Plasmodium falciparum*, one of the parasites that cause malaria. The nopyl-quinolin-8-yl amides (**2**–**4**) were moderately active against the asexual blood stage of chloroquine-sensitive strain *Pf*3D7 but inactive against chloroquine-resistant strains *Pf*K1 and *Pf*NF54. The nopyl-quinolin-4-yl amides and nopyl-quinolin-4-yl-acetates analogs were generally less active on all three strains. Interesting, the presence of a chloro substituent at C7 of the quinoline ring of amide **8** resulted in sub-micromolar EC_50_ in the *Pf*K1 strain. However, **8** was more than two orders of magnitude less active against *Pf*3D7 and *Pf*NF54. Overall, the nopyl-quinolin-8-yl amides appear to share similar antimalarial profile (asexual blood-stage) with previously reported 8-aminoquinolines like primaquine. Future work will focus on investigating the moderately active and selective nopyl-quinolin-8-yl amides on the gametocyte or liver stages of *Plasmodium falciparum* and *Plasmodium vivax*.

## 1. Introduction

Malaria is endemic in 91 countries, which accounts for 40% of the world’s population. According to the US Centers for Disease Control and Prevention, an estimated 228 million malaria cases occurred worldwide in 2018, and 405,000 people died [1]. It remains a serious public health problem in many developing tropical countries. There is no approved vaccine to prevent malaria for the general population. However, there are several therapies whose effectiveness is limited by the strain of *Plasmodium* responsible for the disease or by drug resistance. Artemisinin-based combination therapies (ACTs) are the first-line drugs for the treatment of malaria caused by *P. falciparum* in most malaria-endemic countries, while chloroquine (CQ) or ACT is used for the treatment of uncomplicated *P. vivax* in countries endemic for vivax malaria. Combinations of primaquine with CQ or ACTs are used for radical cure of relapsing malaria parasites due to *P. vivax* and *P. ovale* [2]. The 8-aminoquinoline-based drug primaquine (PQ) is also used as a gametocytocidal agent to reduce the transmission of *P. falciparum* in areas with verified artemisinin resistance [3]. 4-aminoquinoline-based compounds like chloroquine are known for their ability to accumulate as dicationic molecules in the plasmodial food vacuole and inhibit heme detoxification [4,5,6]. Since malarial parasites lack the oxygenase required to metabolize heme, soluble heme (iron-protoporphyrin IX) is converted into an insoluble and inert form called hemozoin, which the parasite can efficiently excrete. When these quinoline-based antimalarial drugs interfere with the conversion of soluble heme to hemozoin, heme accumulates and becomes toxic to the parasite because of radical reactions, damaged cell membranes, and cellular proteopathy [7,8]. The development of resistance to 4-aminoquinoline-based compounds such as CQ is linked to mutations in CQ resistance transporter (*PfCRT*) in *P. falciparum* and upregulation in the expression of CQ resistance transporter (*PvCRT)* in *P. vivax.* Mutations in *P. falciparum* multidrug-resistant protein 1 have also been linked to resistance to artemisinin, halofantrine, mefloquine, and quinine. The mutations and/or upregulated expression shield the parasites from drug accumulation and are viewed as consequences of evolutionary pressure to increase the rate of removal of drugs from the compartment containing the putative drug target(s) [9,10,11,12]. The mechanism of action of 8-aminoquinoline-based antimalarials is unclear. However, some studies have suggested that it involves the generation of reactive oxygen species as well as inhibition of the electron transport chain because of metabolic transformation [13,14]. Quinoline-based compounds remain a robust class of molecules for new antimalarial agents. In this study, quinoline compounds that possess an *α*-pinene scaffold were synthesized (Scheme 1) and evaluated for their antiplasmodial activities. *α*-pinene is a common constituent of essential oils, and it was previously shown to have antiplasmodial activity [15]. Also, phloroglucinol-terpenes robustadial A and B from *Eucalyptus robusta,* containing an *α*-pinene moiety, have been reported to have antiplasmodial activities [16].

## 2. Results and Discussion

The synthetic route to the compounds is shown in Scheme 1 below. To obtain amides **1**–**6** and esters **7**–**12**, (1*R*)-(−)-nopol (**N1**) was oxidized to its corresponding acid using Jones reagent, and the acid was coupled with the corresponding aminoquinolines and quinolinols. Oxidation of nopol produced a relatively low yield of *nopoic acid* (**N2**). This is likely due to cationic rearrangement of the pinene ring to relieve angle strain caused by the four-membered ring bridgehead [17]. The initial reaction of nopoic acid with 5-methoxy-8-aminoquinoline and 8-aminoquinoline, using HBTU-HOBT as coupling reagents, produced two stereoisomers each (**1**–**4**). The diastereotopic *α*-protons in **1** (δ 3.32 and 3.13 ppm) and **3** (δ 3.34 and 3.16 ppm) share similar magnetic fields compared to the diastereotopic *α*-protons in **2** (δ 3.56–3.48 and 2.88 ppm) and **4** (δ 3.55–3.48 and 2.92–2.82 ppm). The observed anisotropic effect is likely due to two different orientations of the quinoline ring in the two sets of isomers. The amides (**1**–**4**) were subsequently tested for their antiplasmodial activities in vitro using the *P. falciparum* 3D7 (*Pf*3D7, chloroquine-sensitive strain), and all four compounds showed moderate but selective low micromolar potency against the parasite (Table 1). The bioactivity (EC_50_) of the compounds was similar to reported EC_50_ values for 8-aminoquinolines like primaquine [18]. This was particularly interesting because the compounds lack the essential side-chain amino moiety typical of quinoline-based antimalarials [19]. Therefore, to extend the compound’s series and to investigate the potentially more potent 4-aminoquinoline derivatives, a set of 4-aminoquinoline-based analogs (**5**–**12**) were synthesized and evaluated against *Pf*3D7. Amides **5** and **6** were weakly active (EC_50_ 26.4 and 17.1 µM, respectively) against *Pf*3D7, and **5** was significantly more cytotoxic to HepG2 (hepatocarcinoma) cells compared to the corresponding 8-aminoquinolyl analogs (**3** and **4**, Table 1). Esters 7–12 were generally inactive against *Pf*3D7 apart from **9** (EC_50_ 6.2 µM). Three other esters, **13**–**15**, were synthesized using 7-chloro-2-methyl-4-quinolinecarboxylic acid, 2-chloro-4-quinolinecarboxylic acid, and 2-indolecarboxylic acid, respectively. Compound **13** displayed very weak antiplasmodial activity while **14** and **15** were moderately active against *Pf*3D7. The 4-aminoquinoline nopyl amides and esters displayed much weaker activities against *Pf*3D7 than the 8-aminoquinoline-derived compounds. To determine if the compounds are active against strains of *P. falciparum* that are less sensitive to chloroquine, compounds **3**–**8** and **14** were assayed against the NF54 and K1 strains of *P. falciparum*. Compounds **3**, **4**, and **14** were inactive against both strains, while **5** and **6** were weakly active against both strains (Table 2). Surprisingly, compound **8**, which bears the 7-chloroquinolyl moiety, was significantly more active against the “multidrug-resistant” K1 strain than the NF54 and 3D7 strains (EC_50 *PfK1*_ 0.16 µM vs. EC_50 *NF54*_ 23.3 µM and EC_50 *Pf3D7*_ > 50 µM). It is unclear why the K1 was more susceptible to **8**. However, a previous report has shown that mutations in CQ resistance transporter, like in the K1 strain, can enhance antimalarial agents’ antiplasmodial activities against CQ-resistant strains [20]. For example, the antiviral agent amantadine can accumulate in the cytosol, act on its molecular target, and exert more significant antiplasmodial activity because the 76I-*Pf*CRT^K1^ transporter variant can transport it back into the cytosol from the food vacuole more effectively than the wild-type transporter (PfCRT^K1^). It is likely that **8** is transported with greater efficiency by *Pf*CRT in the K1 strain and able to engage effectively with a putative cytosolic target, or its transportation out of the food vacuole is significantly hindered in the K1 strain and therefore displayed a more substantial anti-hemozoin effect in the parasite.

Synthetic efforts to introduce epoxy and vicinal diol groups to increase aqueous solubility via the unsaturation in the nopol substructure for **1**–**15** were unsuccessful [21,22,23]. However, **16** was synthesized via epoxidation of nopol followed by esterification reaction with 7-chloro-2-methyl-4-quinolinecarboxylic acid using EDCI and DMAP. Compound **16** was weakly active but nonselective against *Pf*3D7.

In conclusion, the 8-aminoquinolyl amides of nopol were found to be moderately active against the chloroquine-sensitive strain of *P. falciparum* (*Pf*3D7), while the corresponding 4-aminoquinolyl analogs were generally more active on the multidrug-resistant K1 strain, especially ester **8**. Future efforts will be directed towards (1) evaluating the antiplasmodial activity of the 8-aminoquinolyl amides on different life stages of *P. falciparum*, as well as on *P. vivax* and *P. ovale*, and (2) adding an amino functionality to the *α*-pinene or another similar scaffold to potentially enhance the antiplasmodial activities and to understand further the structure–activity relationships of compounds **1**–**4**, **8**, **9**, and **14**.

## 3. Materials and Methods

### 3.1. Compound Characterization

All reagents were purchased from commercial sources and used without further purification. The ^1^H and ^13^C-NMR data were obtained Varian 500 MHz (Agilent, Santa Clara, CA, USA) and Bruker Ultrashield Avance 400 (Bruker, Billerica, MA, USA) spectrometers. Thin layer chromatography (TLC) and NMR were used to monitor reactions and determine compound purity. Compounds were purified by silica gel (Sortech, Norcross, GA, USA, 60 Å, 200–500 µm (35 × 70 mesh)) column chromatography or on pre-coated preparative TLC plates (SiliCycle Inc, Quebec City, Canada, 1000 μm, 20 × 20 cm). High-resolution mass spectrometric (HRMS) data was obtained on a Synapt G2 HDMS instrument operated in positive or negative ESI mode. Spectra are provided as supplementary information (Appendix A).

### 3.2. Synthesis of (R)-(−)-Nopoic Acid Using Jones Reagent

(*R*)-(−)-Nopol (2.2 g, Sigma-Aldrich, St. Louis, MO, USA, 98% purity) was placed in a round bottom flask containing a magnetic stirring bar, and 20 mL of acetone was added. The solution was stirred in an ice-water bath for 10 min. To the solution, 10 mL of 3.88 M Jones reagent (Made from Fisher Scientific’s Chromium(VI) oxide, 99+% purity)in acetone was slowly added for 1 h. TLC was used to monitor the reaction progress. Once no more nopol was detected on analytical TLC, 1–2 mL of isopropanol was added to consume excess Jones reagent. The reaction turned brownish with the subsequent formation of dark green precipitate at the bottom of the flask. The reaction was washed with 20 mL of saturated NaHCO_3_ solution, and the crude product was extracted with 45 mL of ethyl acetate (EtOAc). The EtOAc extract was washed with brine and dried over MgSO_4_ and concentrated under vacuum. Pure nopoic acid was obtained by column chromatography using 85:1 CHCl_3_/ethyl acetate (R_f_ = 0.32). Pure nopoic acid was obtained in 30% yield. ^1^H-NMR (500 MHz, CDCl_3_) δ 11.48 (s, 1H, COOH), 5.42 (s, 1H, H-3′), 3.01 (dd, *J* = 15.7, 15.2 Hz, 2H, H-10′), 2.38 (m, 1H, H-6′), 2.33–2.19 (m, 2H, H-4′), 2.14 (td, *J* = 5.6, 1.6 Hz, 1H, H-5′), 2.09 (m, 1H, H-1′), 1.28 (s, 3H, H-9′), 1.22 (d, *J* = 8.7 Hz, 1H, H-6′), 0.83 (s, 3H, H-8′).^13^C-NMR (126 MHz, CDCl_3_) δ 178.0 (C=O), 140.5 (C-2′), 121.3 (C-3′), 45.9 (C-10′), 42.4 (C-5′), 40.5 (C-1′), 38.1 (C-7′), 31.7 (C-6′), 31.5 (C-4′), 26.2 (C-9′), 20.9 (C-8′).

### 3.3. Synthesis of (2-(6,6-dimethylbicyclo[3.1.1]hept-2-en-2-yl)-N-(5-methoxyquinolin-8-yl Acetamide (***1*** and ***2***)

To a round bottom flask containing 3 mL of *N*,*N*-dimethylformamide (DMF, Fisher Scientific, >99.8%), nopoic acid (200 mg, 1.1 mmol, 1.0 eq.) was added, followed by 0.1 mL of *N*,*N*-diisopropylethylamine (DIEA, Sigma-Aldrich, 99.5% purity), *N*,*N*,*N*′,*N*′-Tetramethyl-*O*-(1*H*-benzotriazol-1-yl)uronium hexafluorophosphate (HBTU, Sigma-Aldrich, 98% purity, 400.3 mg, 0.1 mmol), and HOBT (1-Hydroxybenzotriazole hydrate, 98% purity, 144.74 mg, 1.1 mmol, 1.0 eq.) [24]. The reaction mixture was stirred at room temperature for 30 min, and 5-methoxy-8-aminoquinoline (186.3 mg, 1.1 mmol, and 1.0 eq, Sigma-Aldrich, 95% purity) was added to it. The reaction was allowed to run for 16 h. TLC results show some unreacted amine and activated nopoic acid with two prominent product spots. The compounds (**1** and **2**) were purified using preparative TLC (10:1 hexane/ethyl acetate, R_f_ = 0.55 (**1**) and 0.68 (**2**)), and the overall product yield was 35%.

#### 3.3.1. Compound **1**



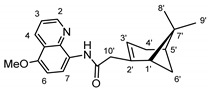



^1^H-NMR (500 MHz, CDCl_3_) δ 9.81 (s, 1H, NH), 8.79 (dd, J = 4.2, 1.7 Hz, 1H, H-2), 8.68 (d, *J* = 8.6 Hz, 1H, H-6), 8.56 (dd, *J* = 8.4 Hz, *J* = 1.7 Hz, 1H, H-4), 7.42 (dd, *J* = 8.4 Hz, *J* = 4.2 Hz, 1H, H-3), 6.83 (d, *J* = 8.6 Hz, 1H, H-7), 5.66 (m, 1H, H-3′), 3.98 (s, 3H, OMe), 3.32 (d, *J* = 14.2 Hz, 1H, H-10′), 3.13 (m, 1H, H-10′), 2.43–2.39 (m, 2H, H-6′, H-4′), 2.35–2.29 (m, 1H, H-1′), 2.22 (m, 1H, H-5′), 2.13 (m, 1H, H-4′), 1.48 (d, *J* = 8.7 Hz, 1H, H-6′), 1.26 (s, 3H, H-9′), 0.85 (s, 3H, H-8′). ^13^C-NMR (126 MHz, CDCl_3_) δ 169.1 (C=O), 150.1 (C-5), 148.6 (C-2), 142.2 (C-2′), 139.2 (C-8), 131.1 (C-6), 128.1 (C-8a), 122.3 (C7), 120.6 (C-3), 120.4 (C-4a), 116.4 (C-3′), 104.3 (C-4), 55.8 (OMe), 46.9 (1′), 45.9 (C-10′), 40.4 (C-5′), 38.1 (C-6′), 31.9 (C-7′), 31.6 (C-4′), 26.1 (C-9′), 20.9 (C-8′). HRMS: [M + H]^+^: 337.1916 m/z calculated for C_21_H_25_N_2_O_2_, found 337.1910 *m*/*z*.

#### 3.3.2. Compound **2**



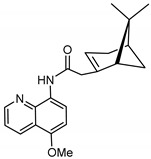



^1^H-NMR (500 MHz, CDCl_3_) δ 9.49 (s, 1H, NH), 8.79 (dd, *J* = 4.3, 1.9 Hz, 1H, H-2), 8.76 (d, *J* = 8.6 Hz, 1H, H-6), 8.56 (dd, *J* = 8.5, 1.8 Hz, 1H, H-4), 7.43 (dd, *J* = 8.3, 4.2 Hz, 1H, H-3), 6.84 (d, *J* = 8.6 Hz, 1H, H-7), 5.76 (d, *J* = 2.2 Hz, 1H, H-3′), 3.98 (s, 3H, OMe), 3.52 (m, 1H, H-10′), 2.88 (m, 1H, H-10′), 2.54 (m, 1H, H-6′), 2.41 (m, 1H, H-4′), 2.09 (m, 1H, H-1′), 2.01-1.88 (m, 2H, H-5′, H4′), 1.46 (d, *J* = 10.0 Hz, 1H, H-6′), 1.29 (s, 3H, H-9′), 0.80 (s, 3H, H-8′). ^13^C-NMR (126 MHz, CDCl_3_) δ 165.2 (C=O), 149.8 (C-5), 148.5 (C-2), 139.2 (C-2′), 131.3 (C-6), 128.7 (C-8), 120.7 (C-7), 120.6 (C-8a), 117.2 (C-4a), 116.2 (C-3′), 116.0 (C-3), 104.6 (C-4), 55.9 (OMe), 53.9 (C-1′), 41.1 (C-10′), 40.7 (C-5′), 27.4 (C-6′), 26.3 (C-9′), 23.9 (C-7′), 22.4 (C-4′), 22.30 (C-8′). HRMS: [M + H]^+^: 337.1916 m/z calculated for C_21_H_25_N_2_O_2_, found 337.1916 m/z.

### 3.4. Synthesis of 2-(6,6-dimethylbicyclo[3.1.1]hept-2-en-2-yl)-N-(quinolin-8-yl) Acetamide (***3*** and ***4***)

Compounds **3** and **4** were synthesized as described for **1** and **2** above and separated by preparative TLC (85:1 CHCl_3_/ethyl acetate, R_f_ = 0.81 (**3**) and 0.85 (**4**)) with 25% yield. 8-Aminoquinoline was obtained from Oakwood Chemicals, Estill, SC, USA. 

#### 3.4.1. Compound **3**



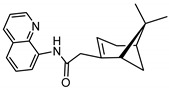



^1^H-NMR (500 MHz, CDCl_3_) δ 10.06 (s, 1H, NH), 8.78 (m, 1H, H-2), 8.76 (d, *J* = 7.8 Hz, 1H, H-4), 8.14 (dd, *J* = 8.2, 1.7 Hz, 1H, H-5), 7.53 (t, *J* = 7.8 Hz, 1H, H-6), 7.48 (dd, *J* = 8.3, 1.6 Hz, 1H, H-7), 7.43 (dd, *J* = 8.3, 4.2 Hz, 1H, H-3), 5.67 (s, 1H, H-3′), 3.34 (m, 1H, H-10′), 3.16 (dd, *J* = 15.3, 1.8 Hz, 1H, H-10′), 2.45 (m, 1H, H-4′), 2.40 (m, 1H, H-6′), 2.32 (m, 1H, H-4′), 2.22 (d, *J* = 5.6 Hz, 1H, H-1′), 2.12 (s, 1H, H-5′), 1.48 (d, *J* = 8.8 Hz, 1H, H-6′), 1.26 (s, 3H, H-9′), 0.85 (s, 3H, H-8′). ^13^C-NMR (126 MHz, CDCl_3_) δ 169.6 (C=O), 148.3 (C-2), 142.2 (C-2′), 138.6 (C-8), 136.4 (C-5), 134.7 (C-8a), 128.0 (C-4a), 127.7 (C-6), 122.6 (C-7), 121.7 (C-3), 121.5 (C-4), 116.5 (C-3′), 47.2 (C-1′), 46.0 (C-10′), 40.5 (C-5′), 38.3 (C-6′), 32.1 (C-4′), 31.8 (C-7′), 26.2 (C-9′), 21.1 (C-8′). HRMS: [M + H]^+^: 307.1810 *m*/*z* calculated for C_20_H_23_N_2_O, found 307.1804 *m*/*z*.

#### 3.4.2. Compound **4**



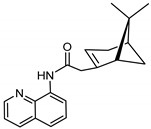



^1^H-NMR (500 MHz, CDCl_3_) δ 9.74 (s, 1H, NH), 8.85 (d, *J* = 7.8 Hz, 1H, H-2), 8.80 (dd, *J* = 4.3, 1.7 Hz, 1H, H-4), 8.16 (dd, *J* = 8.3, 1.7 Hz, 1H, H-5), 7.54 (t, *J* = 7.9 Hz, 1H, H-6), 7.49-7.47 (m, 1H, H-7), 7.45 (dd, *J* = 8.2, 4.2 Hz, 1H, H-3), 5.80 (s, 1H, H-3′), 3.55–3.48 (m, 1H, H-10′), 2.92–2.82 (m, 1H, H-10′), 2.57 (m, 1H, H-6′), 2.42 (m, 1H, H-4′), 2.09 (m, 1H, H-1′), 1.99 (m, 1H, H-5′), 1.93 (m, 1H, H-4′), 1.46 (d, *J* = 9.9 Hz, 1H, H-6′), 1.29 (s, 3H, H-9′), 0.81 (s, 3H, H-8′). ^13^C-NMR (126 MHz, CDCl_3_) δ 166.4 (C=O), 165.5 (C-2), 148.1 (C-2′), 138.6 (C-8), 136.5 (C-5), 135.3 (C-8a), 128.1 (C-4a), 127.7 (C-6), 121.6 (C-7), 121.1 (C-3), 116.1 (C-3′), 115.9 (C-4), 54.0 (C-1′), 41.2 (C-5′), 40.7 (C-10′), 27.4 (C-6′), 26.3 (C-9′), 23.9 (C-7′), 22.5 (C-4′), 22.3 (C-8′). HRMS: [M + H]^+^: 307.1810 *m*/*z* calculated for C_20_H_23_N_2_O, found 307.1803 *m*/*z*.

### 3.5. Synthesis of 2-((1S,5R)-6,6-dimethylbicyclo[3.1.1]hept-2-en-2-yl)-N-(quinolin-4-yl) Acetamide (***5***)

Compound **5** was synthesized as described for **1** and **2** above and separated by preparative TLC (20:1 CHCl_3_/ethyl acetate, R_f_ = 0.55) and 25% yield. 4-Aminoquinoline (95% purity) was obtained from Ark Pharm, Inc., Arlington Heights, IL, USA.



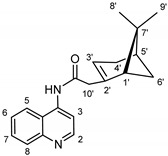



^1^H-NMR (500 MHz, CDCl_3_) δ 8.85 (d, *J* = 5.2 Hz, 1H, H-2), 8.34 (d, *J* = 5.2 Hz, 1H, H-3), 8.14 (d, *J* = 8.4 Hz, 1H, H-5), 7.74 (m, 1H, H-6), 7.73-7.69 (m, 1H, H-8), 7.58 (t, *J* = 7.58 Hz, 1H, H-7), 5.77 (s, 1H, H-3′), 3.37 (d, *J* = 15.5 Hz, 1H, H-10′), 3.18 (d, *J* = 15.4 Hz, 1H, H-10′), 2.53–2.48 (m, 1H, H-6′), 2.47 (m, 1H, H-5′), 2.42–2.36 (m, 1H, H-4′), 2.25–2.16 (m, 2H, H-4′), 2.19 (s, 1H, H-6′), 1.29 (s, 3H, H-9′), 1.26 (m, 1H, H-6′), 0.86 (s, 3H, H-8′). ^13^C-NMR (126 MHz, CDCl_3_) δ 169.4 (C=O), 151.2 (C-2), 142.7 (C-2′), 140.3 (C-4a), 130.5 (C-8a), 129.4 (C-5), 126.6 (C-6), 123.6 (C-7), 119.7 (C-3), 118.6 (C-3′), 113.3 (C-8), 110.3 (C-4), 47.0 (C-1′), 45.8 (C-10′), 40.3 (C-5′), 38.2 (C-6′), 32.5 (C-4′), 31.7 (C-7′), 26.1 (C-9′), 21.1 (C-8′). HRMS: [M − H]^-^: 305.1654 *m*/*z* calculated for C_20_H_21_N_2_O, found 305.1655 *m*/*z*.

### 3.6. Synthesis of N-(7-chloroquinolin-4-yl)-2-((1S,5R)-6,6-dimethylbicyclo[3.1.1]hept-2-en-2-yl) Acetamide (***6***)

Compound **6** was synthesized as described for **1** and **2** above and separated by preparative TLC (20:1 hexane/ethyl acetate, R_f_ = 0.78) and 36% yield. 7-Chloro-4-aminoquinoline (95% purity) was obtained from Life Chemicals, Ontario, Canada.



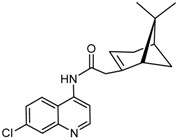



^1^H-NMR (500 MHz, CDCl_3_) δ 8.83 (d, *J* = 5.1 Hz, 1H, H-2), 8.31 (d, *J* = 5.2 Hz, 1H, H-3), 8.11 (s, *J* = 2.1 Hz, 1H, H-8), 7.65 (d, *J* = 9.0 Hz, 1H, H-5), 7.52 (dd, *J* = 9.0, 2.1 Hz, 1H, H-6), 5.79–5.75 (m, 1H, H-3′), 3.37 (d, *J* = 16.6 Hz, 1H, H-10′), 3.18 (d, *J* = 17.5 Hz, 1H, H-10′), 2.54–2.49 (m, 1H, H-6′), 2.49–2.44 (m, 1H, H-5′), 2.38 (m, 1H, H-1′), 2.21 (m, 2H, H-4′), 1.29 (s, 3H, H-9′), 1.23 (d, *J* = 8.8 Hz, 1H, H-6′), 0.86 (s, 3H, H-8′). ^13^C-NMR (126 MHz, CDCl_3_) δ 169.5 (C=O), 152.4 (C-2), 142.8 (C-2′), 138.7 (C-4a) 136.7 (C-7), 135.6 (C-8), 129.5 (C-8a), 127.6 (C-5), 123.9 (C-6), 120.5 (C-3), 118.3 (C-3′), 110.7 (C-4), 47.1 (C-1′), 45.9 (C-10′), 40.4 (C-5′), 38.3 (C-6′), 32.7 (C-4′), 31.9 (C-7′), 26.1 (C-9′), 21.2 (C-8′). HRMS: [M − H]^−^: 339.1264 *m*/*z* calculated for C_20_H_20_ClN_2_O, found 339.1263 *m*/*z*.

### 3.7. Synthesis of quinolin-4-yl-2-((1S,5R)-6,6-dimethylbicyclo[3.1.1]hept-2-en-2-yl) Acetate (***7***)



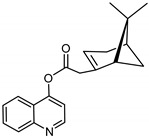



*N*-(3-Dimethylaminopropyl)-*N*′-ethylcarbodiimide hydrochloride (EDCI, Sigma-Aldrich, 99% purity, 171.7 mg, 0.90 mmol, 1.3 eq.), DMAP (4-Dimethylaminopyridine, 99%, ACROS Organics™, 98.5% purity, 25.7 mg, 0.21 mmol, 0.5 eq.), and DIEA (2.04 mmol, 0.36 mL, and 3eq.) were added to a solution of 4-hydroxyquinoline (100 mg, 0.68 mmol, 1.0 eq., Ark Pharm, Inc., 97% purity) and nopoic acid (0.42 mmol, 1.2 eq.) in tetrahydrofuran (THF, Fisher Scientific, 99.9%, 2.0 mL) at 0 °C and stirred for 15 min. [25]. The temperature was raised to 23 °C, and the reaction mixture was stirred for 3 h while monitoring by analytical TLC. The reaction appeared cloudy at first but after stirring for 1 h at room temperature, it became clear, with the formation of colorless solid precipitate. The reaction mixture was quenched with brine, extracted with EtOAc, concentrated, and separated by preparative TLC (20:1 CHCl_3_/ethyl acetate, R_f_ = 0.55) with 25% yield.

^1^H-NMR (500 MHz, CD_3_OD) δ 8.84 (d, *J* = 5.0 Hz, 1H, H-2), 8.05 (d, *J* = 8.5 Hz, 1H, H-8), 8.02 (d, *J* = 8.5, 1.4 Hz, 1H, H-5), 7.81 (ddd, *J* = 8.4, 6.9, 1.4 Hz, 1H, H-6), 7.64 (ddd, *J* = 8.2, 6.8, 1.2 Hz, 1H, H-7), 7.44 (d, *J* = 5.0 Hz, 1H, H-3), 5.97 (m, 1H, H-3′), 3.34–3.28 (m, 1H, H-10′), 2.82-2.70 (m, 2H, H-10′, H-1′), 2.52 (m, 1H, H-6′), 2.11 (m, 1H, H-5′), 2.02 (m, 1H, H-4′), 1.96 (m, 1H, H-4′), 1.48 (d, *J* = 10.2 Hz, 1H, H-6′), 1.34 (s, 3H, H-9′), 0.84 (s, 3H, H-8′). ^13^C-NMR (126 MHz, CD_3_OD) δ 177.2 (C=O), 164.3 (C-4a), 156.5 (C-2), 151.9 (C-7), 150.5 (C-2), 131.7 (C-8), 129.2 (C-8a), 124.2 (C-6), 122.8 (C-3), 114.6 (C-3′), 111.4 (C-4′), 42.1 (C-1′), 41.6 (C-10′), 34.7 (C-5′), 28.0 (C-6′), 26.7 (C-4′), 26.4 (C-7′), 26.1 (C-9′), 24.5 (C-8′). [M + H]^+^: 308.1650 *m*/*z* calculated for C_20_H_22_NO_2_, found 308.1651 *m*/*z*.

### 3.8. Synthesis of 7-chloroquinolin-4-yl-2-((1S,5R)-6,6-dimethylbicyclo[3.1.1]hept-2-en-2-yl) Acetate (***8***)

Compound **8** was synthesized as described for **7** above and separated by preparative TLC (20:1 hexane/ethyl acetate and R_f_ = 0.85) with 40% yield. 7-Chloro-4-hydroxyquinoline (99% purity) was obtained from Sigma-Aldrich.



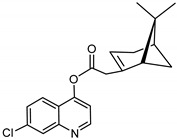



^1^H-NMR (500 MHz, (CD_3_)_2_CO) δ 8.95 (d, *J* = 4.8 Hz, 1H, H-2), 8.10 (d, *J* = 2.2 Hz, 1H, H-8), 8.06 (d, *J* = 9.0 Hz, 1H, H-5), 7.62 (dd, *J* = 8.9, 2.0 Hz, 1H, H-6), 7.49 (d, *J* = 4.9 Hz, 1H, H-3), 5.99 (m, 1H, H-3′), 3.31 (dd, *J* = 20.4, 9.2 Hz, 1H, H-10′), 2.84–2.78 (m, 1H, H-10′), 2.75 (m, 1H, H-1′), 2.51 (dt, *J* = 10.8, 5.8 Hz, 1H, H-6′), 2.14–2.10 (m, 1H, H-5′), 2.02 (ddt, *J* = 10.1, 3.9, 1.9 Hz, 1H, H-4′), 1.94 (m, 1H, H-4′), 1.49 (d, *J* = 10.1 Hz, 1H, H-6′), 1.33 (s, 3H, H-9′), 0.84 (s, 3H, H-8′). ^13^C-NMR (126 MHz, (CD_3_)_2_CO) δ 176.1 (C=O), 163.4 (C-4a), 155.3 (C-2), 153.4 (C-7), 151.2 (C-2′), 136.1 (C-8), 129.0 (C-8a), 128.3 (C-5), 124.5 (C-6), 122.2 (C-3), 114.6 (C-3′), 111.3 (C-4), 54.7 (C-1′), 41.7 (C-10′), 41.1(C-5′), 27.6 (C-6′), 26.2 (C-9′), 24.1 (C-4′), 23.6 (C-7′), 22.4 (C-8′). [M + H]^+^: 342.1261 *m*/*z* for calculated C_20_H_21_ClNO_2_, found 342.1259 *m*/*z*.

### 3.9. Synthesis of 7-chloro-2,8-dimethylquinolin-4-yl-2-((1S,5R)-6,6-dimethylbicyclo[3.1.1]hept-2-en-2-yl) Acetate (***9***)

Compound **9** was synthesized as described for **7** above and separated by preparative TLC (4:1 hexane/ethyl acetate, R_f_ = 0.89) with 39% yield. 7-Chloro-2,8-dimethyl-4-hydroxyquinoline (98% purity) was obtained from Sigma-Aldrich.



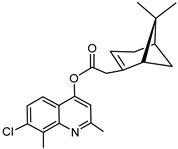



^1^H-NMR (500 MHz, (CD_3_)_2_CO) δ 7.78 (d, *J* = 9.0 Hz, 1H, H-6), 7.50 (d, *J* = 8.9 Hz, 1H, H-5), 7.31 (s, 1H, H-3), 5.96 (m, 1H, H-3′), 3.34–3.27 (m, 1H, H-10′), 2.82 (s, 3H, C-3Me), 2.77 (m, 1H, H-10′) 2.74 (m, 1H, H-1′), 2.71 (s, 3H, C-8Me), 2.51 (m, 1H, H-6′), 2.11 (m, 1H, H-5′), 2.03–1.99 (m, 1H, H-4′), 1.97–1.91 (m, 1H, H-4′), 1.48 (d, *J* = 10.1 Hz, 1H, H-6′), 1.33 (s, 3H, H-9′), 0.83 (s, 3H, H-8′). ^13^C-NMR (126 MHz (CD_3_)_2_CO) δ 175.6 (C=O), 163.5 (C-4a), 160.8 (C-2), 155.6 (C-7), 149.6 (C-2′), 135.6 (C-8), 134.9 (C-8a), 127.6 (C-5), 120.9 (C-6), 120.5 (C-3), 114.6 (C-3′), 111.4 (C-4), 54.7 (C-1′), 41.7 (C-10′), 41.1 (C-5′), 27.6 (C-6′), 26.2 (C-9′), 25.8 (C-4′), 24.1 (C-2Me), 23.5 (C-7′), 22.4 (C-8′), 14.7 (C-8Me). [M + H]^+^: 370.1574 *m*/*z* for calculated C_22_H_25_ClNO_2,_ found 370.1574 *m*/*z*.

### 3.10. Synthesis of 6-fluoro-2-(trifluoromethyl)quinolin-4-yl-2-((1S,5R)-6,6-dimethylbicyclo[3.1.1]hept-2-en-2yl) Acetate (***10***)

Compound **10** was synthesized as described for **7** above and separated by preparative TLC (4.5:1 hexane/ethyl acetate, R_f_ = 0.78) with 33% yield. 6-Fluoro-4-hydroxy-2-(trifluoromethyl)quinoline (98% purity) was obtained from Sigma-Aldrich.



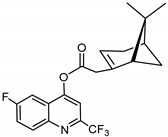



^1^H-NMR (500 MHz, CDCl_3_) δ 8.25 (m, 1H, H-7), 7.78 (s, 1H, H-3), 7.67 (m, 1H, H-8), 7.60 (s, 1H, H-5), 5.91 (s, 1H, H-3′), 3.33 (dd, *J* = 20.5, 9.3 Hz, 1H, H-10′), 2.88–2.73 (m, 1H, H-10′), 2.71 (t, *J* = 5.4 Hz, 1H, H-1′), 2.50 (m, 1H, H-6′), 2.16 (m, 1H, H-5′), 2.05–1.93 (m, 2H, H-4′), 1.47 (d, *J* = 10.2 Hz, 1H, H-6′), 1.34 (s, 3H, H-9′), 0.85 (s, 3H, H-8′). ^13^C-NMR (126 MHz, CDCl_3_) δ 177.7 (C=O), 162.4 (C-6), 146.2 (C-2), 142.2 (C-2′), 133.1 (C-8), 133.0 (C-4a), 124.2 (CF_3_), 121.9 (C-7), 121.7 (C-8a), 110.1 (C-3′), 109.8 (C-5), 105.8 (C-3), 105.6 (C-4), 54.4 (C-1′), 41.4 (C-10′), 40.4 (C-5′), 27.4 (C-6′), 26.2 (C-9′), 23.7 (C-4′), 23.4 (C-7′), 22.4 (C-8′). [M + Na]^+^: 416.1250 *m*/*z* calculated for C_21_H_19_F_4_NO_2_Na, found: 416.1245 *m*/*z*.

### 3.11. Synthesis of 7-chloro-8-methylquinolin-4-yl-2-((1S,5R)-6,6-dimethylbicyclo[3.1.1]hept-2-en-2-yl) Acetate (***11***)

Compound **11** was synthesized as described for **7** above and separated by preparative TLC (5:1 hexane/ethyl acetate, R_f_ = 0.82) with 38% yield. 7-Chloro-4-hydroxy-8-methylquinoline (98% purity) was obtained from Sigma-Aldrich.



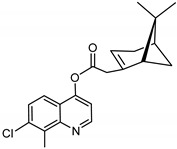



^1^H-NMR (500 MHz, CDCl_3_) δ 8.92 (d, *J* = 4.9 Hz, 1H, H-2), 7.79 (d, *J* = 8.9 Hz, 1H, H-6), 7.51 (d, *J* = 9.0 Hz, 1H, H-5), 7.33 (d, *J* = 4.9 Hz, 1H, H-3), 5.90 (t, *J =* 2.3 Hz, 1H, H-3′), 3.36–3.27 (m, 1H, H-10′), 2.88 (s, 3H, C-8Me), 2.76 (m, 1H, H-10′), 2.68 (t, *J* = 5.3 Hz, 1H, H-1′), 2.48 (m, 1H, H-6′), 2.13 (s, 1H, H-5′), 2.03-1.98 (m, 1H, H-4′), 1.93 (m, 1H, H-4′), 1.46 (d, *J* = 10.1 Hz, 1H, H-6′), 1.25 (s, 3H, H-9′), 0.84 (s, 3H, H-8′). ^13^C-NMR (126 MHz, CDCl_3_) δ 175.8 (C=O), 163.0 (C-4a), 154.6 (C-2), 150.5 (C-7), 149.8 (C-2′), 135.4 (C-8), 135.0 (C-8a), 128.0 (C-5), 121.0 (C-6), 119.9 (C-3), 112.8 (C-3′), 110.5 (C-4), 54.2 (C-1′), 41.1 (C-10′), 40.35 (C-5′), 31.92 (C-7′), 27.24 (C-6′), 26.02 (C-9′), 23.53 (C-4′), 22.23 (C-8′), 14.77 (C-8Me). [M + Na]^+^: 378.1237 *m*/*z* for calculated for C_21_H_22_ClNO_2_Na found 378.1234 *m*/*z*.

### 3.12. Synthesis of 2,8-bis(trifluoromethyl)quinolin-4-yl-2-((1S,5R)-6,6-dimethylbicyclo[3.1.1]hept-2-en-2-yl) Acetate (***12***)

Compound **12** was synthesized as described for **7** above and separated by preparative TLC (4.5:1 hexane/ethyl acetate, R_f_ = 0.76) with 31% yield. 2,8-Bis(trifluoromethyl)-4-quinolinol (99% purity, Acros Organics) was obtained from Thermo Fisher Scientific, Waltham, MA.



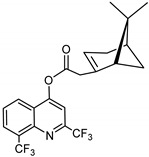



^1^H-NMR (500 MHz, CDCl_3_) δ 8.28 (d, *J* = 8.6 Hz, 1H, H-7), 8.18 (d, *J* = 7.4 Hz, H-5), 7.84 (s, 1H, H-3), 7.74–7.67 (m, 1H, H-6), 5.92 (s, 1H, H-3′), 3.38–3.28 (m, 1H, H-10′), 2.80 (m, 1H, H-10′), 2.71 (m, 1H, H-1′), 2.50 (m, 1H, H-6′), 2.16 (m, 1H, H-5′), 2.08–2.00 (m, 1H, H-4′), 1.96 (m, 1H, H-4′), 1.47 (d, *J* = 10.3 Hz, 1H, H-6′), 1.34 (s, 3H, H-9′), 0.85 (s, 3H, H-8′). ^13^C-NMR (126 MHz, CDCl_3_) δ 178.0 (C=O), 162.4 (C-6), 156.2 (C-2), 149.3 (C-8a), 145.4 (C-2′), 129.6 (C-8), 129.6 (C-4a), 127.21 (C-4), 126.3 (C-5), 124.7 (CF_3_), 124.0 (CF_3_), 122.6 (C-7), 122.5 (C-3), 110.1 (C-3′), 54.5 (C-1′), 41.4 (C-10′), 40.4 (C-5′), 27.4 (C-6′), 26.2 (C-9′), 23.7 (C-4′), 23.5 (C-8′), 22.4 (C-8′). [M + Na]^+^: 466.1218 *m*/*z* calculated for C_22_H_19_F_6_NO_2_Na, found 466.1220 *m*/*z*.

### 3.13. Synthesis of 2-((1S,5R)-6,6-dimethylbicyclo[3.1.1]hept-2-en-2-yl)ethyl-7-chloro-2-methylquinoline-4-carboxylate (***13***)

EDCI (112.1 mg, 0.58 mmol, and 1.3 eq.) was added to a solution of 7-chloro-2 methyl-4-quinolinecarboxylic acid (99.7 mg, 0.45 mmol, 1.0 eq., Sigma-Aldrich, 98% purity) in THF (1.0 mL) at 0 °C and stirred for 15 min. Nopol (90.0 mg, 0.54 mmol, 1.2 eq.) in THF (1.0 mL) was added, followed by DIEA (2.04 mmol, 0.36 mL, 3 eq.). The reaction mixture was stirred at room temperature for 3 h while monitoring by analytical TLC. The reaction mixture was quenched with brine, extracted with EtOAc, concentrated, and separated by preparative TLC (3:1 hexane/ethyl acetate, R_f_ = 0.92) with 35% yield.



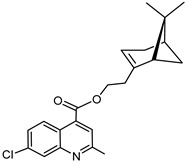



^1^H-NMR (500 MHz, CDCl_3_) δ 8.69 (dd, *J* = 9.1, 1.5 Hz, 1H, H-6), 8.06 (d, *J* = 1.9 Hz 1H, H-8), 7.78 (d, *J* = 1.6 Hz 1H, H-3), 7.51 (d, *J* = 9.1 Hz, 1H, H-5), 5.40 (m, 1H, H-3′), 4.45 (m, 2H, H-11′), 2.77 (d, *J* = 1.7 Hz, 3H, C-2Me), 2.49 (s, 2H, H-10′), 2.43–2.38 (m, 1H, H-6′), 2.33-2.20 (m, 2H, H-4′), 2.15–2.09 (m, 2H, H-5′, H-1′), 1.28 (s, 3H, H-9′), 1.18 (d, *J* = 8.6 Hz, 1H, H-6′), 0.85 (s, 3H, H-8′). ^13^C-NMR (126 MHz, CDCl3) δ 166.1 (C=O), 160.0 (C-3), 149.5 (C-2), 144.0 (C-2′), 135.8 (C-8), 135.3 (C-8a), 128.3 (C-5), 128.2 (C-6), 127.1 (C-4), 123.6 (C-4a), 122.0 (C-3), 119.5 (C-3′), 64.3 (C-11′), 45.9 (C-10′), 40.8 (C-1′), 38.2 (C-5′), 36.1 (C-7′), 31.9 (C-6′), 31.6 (C-4′), 26.4 (C-9′), 25.3 (C-2Me), 21.3 (C-8′). [M + H]^+^: 370.1574 *m*/*z* calculated for C_22_H_25_ClNO_2_, found 370.1575 *m*/*z*.

### 3.14. Synthesis of 2-((1S,5R)-6,6-dimethylbicyclo[3.1.1]hept-2-en-2-yl)ethyl-2-chloroquinoline-3-carboxylate (***14***)

Compound **14** was synthesized as described for **13** above and separated by preparative TLC preparative TLC by preparative TLC (40:1 CHCl_3_/Et_2_O, R_f_ = 0.72) with 50% yield. The acid, 2-chloroquinoline-3-carboxylic acid (97% purity) was obtained from Sigma-Aldrich.



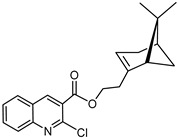



^1^H-NMR (500 MHz, CD_3_OD) δ 8.76 (s, 1H, H-4), 8.01 (d, *J* = 8.1 Hz, 1H, H-5), 7.94 (d, *J* = 8.5 Hz, 1H, H-8), 7.88 (dd, *J* = 8.5, 6.9 Hz, 1H, H-7), 7.68 (t, *J* = 7.6 Hz, 1H, H-6), 5.39 (s, 1H, H-3′), 4.38 (qt, *J* = 10.9, 6.8 Hz, 2H, H-11′), 2.46 (m, 2H, H-10′), 2.40 (dt, *J* = 8.7, 5.7 Hz, 1H, H-6′), 2.30-2.17 (m, 2H, H-4′), 2.17-2.12 (m, 1H, H-5′), 2.06 (m, 1H, H-1′), 1.27 (s, 3H, H-9′), 1.17 (d, *J* = 8.6 Hz, 1H, H-6′), 0.84 (s, 3H, H-8′). ^13^C-NMR (126 MHz, CD_3_OD) 165.70 (C=O), 149.2 (C-2), 148.4 (C-4a), 145.5 (C-3), 143.0 (C-3′), 134.1 (C-6), 129.9 (C-8), 129.3 (C-8a), 128.7 (C-4), 127.3 (C-5), 125.9 (C-7), 120.2 (C-3′), 65.3 (C-11′), 46.9 (C-10′), 42.0 (C-5′), 38.9 (C-1′), 36.9 (C-6′), 32.6 (C-7′), 32.3 (C-4′), 26.7 (C-9′), 21.6 (C-8′). [M + H]^+^: 356.1417 *m*/*z* calculated for C_21_H_23_ClNO_2_, found 356.1409 *m*/*z*.

### 3.15. Synthesis of 2-((1R,5S)-6,6-dimethylbicyclo[3.1.1]hept-2-en-2-yl)ethyl-1H-indole-2-carboxylate (***15***)

Compound **15** was synthesized as described for **13** above and separated by preparative TLC (40:1 CHCl_3_/Et_2_O, R_f_ = 0.72) with 50% yield. Indole 2-carboxylic acid (98% purity) was obtained from Sigma-Aldrich.



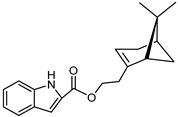



^1^H-NMR (500 MHz, CD_3_OD) δ 7.61 (d, *J* = 8.1 Hz, 1H, H-7), 7.43 (d, *J* = 8.4 Hz, 1H, H-4), 7.24 (t, *J* = 7.7 Hz, 1H, H-5), 7.11 (s, 1H, H-3), 7.06 *(*t, *J* = 7.6 Hz, 1H, H-6), 5.39 (*s*, 1H, H-3′), 4.38–4.27 *(*m, 2H, H-11′), 2.47–2.39 (m, 3H, H-10′, H-6′), 2.31–2.19 (m, 2H, H-4′), 2.17 (t, *J* = 5.6 Hz, 1H, H-5′), 2.07 (s, 1H, H-1′), 1.28 (s, 3H, H-9′), 1.19 (d, *J* = 8.6 Hz, 1H, H-6′), 0.86 (s, 3H, H-8′). ^13^C-NMR (126 MHz, CD_3_OD) δ 163.5 (C=O), 145.8 (C-3′), 139.0 (C-7a), 128.6 (C-3a), 126.0 (C-2), 123.1 (C-4), 121.3 (C-6), 120.2 (C-5), 113.3 (C-2′), 109.2 (C-7), 64.2 (C-11′), 47.0 (C-10′), 42.0 (C-1′), 40.0 (C-5′), 37.1 (C-7′), 32.6 (C-6′), 32.4 (C-4′), 26.7 (C-9′), 24.2 (C-9), 21.6 (C-8) HRMS: [M + H]^+^: 310.1807 *m*/*z* calculated for C_20_H_24_NO_2_, found 310.1806 *m*/*z*.

### 3.16. Synthesis of 2-((1S,2S,4R,6S)-7,7-dimethyl-3-oxatricyclo[4.1.1.0 2,4]octan-2-yl)ethyl-7-chloro-2-methylquinoline-4-carboxylate (***16***)

Nopol (500 mg, 3.0 mmol, 1.eq.) was added to a round 100 mL bottom flask, followed by 4.0 mL of ethyl acetate, 4.0 mL of deionized H_2_O, and 8.0 mL of acetone, and stirred at room temperature. NaHCO_3(s)_ (9.0 mmol, 756 mg, 3.0 eq.) was added to the mixture and stirred for 15 min. Oxone salt (3.6 mmol, 2.4 g, 1.2 eq.), dissolved in 1.0 mL of water, was added dropwise for 1 h, and analytical TLC was used to monitor the reaction for 2.5 h [26]. The reaction was quenched with 3.0 mL of H_2_O. The organic layer was extracted with 15.0 mL CHCl_3_ (3 times), dried over MgSO_4_(s), and concentrated. The crude colorless solid obtained was separated by preparative TLC (10:1 CHCl_3_/ethyl acetate, R_f_ = 0.37) with 40% yield of nopol oxide. EDCI (112.1 mg, 0.58 mmol, 1.3 eq.), DMAP (27.5 mg, 0.23 mmol, 0.5 eq.), and DIEA (2.04 mmol, 0.36 mL, 3 eq.) were added to a solution of 7-chloro-2-methyl-4-quinolinecarboxylic acid (99.7 mg, 0.45 mmol, 1.0 eq., Sigma-Aldrich, 98%) and nopol oxide (151.9 mg, 0.54 mmol, and 1.2 eq.) in THF (2.0 mL) at 0 °C and stirred for 15 min. The temperature was raised to 23 °C, and the mixture was stirred for 3 h while monitoring by analytical TLC, quenched with brine, extracted with EtOAc, and separated by preparative TLC (4.5:1 hexane/ethyl acetate, R_f_ = 0.59) with 49% yield.



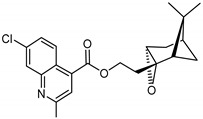



^1^H-NMR (500 MHz, CDCl_3_) δ 8.68 (d, *J* = 9.1 Hz, 1H, H-5), 8.06 (d, *J* = 2.2 Hz, 1H, H-8), 7.79 (s, 1H, H-3), 7.53 (d, *J* = 9.1, 2.2 Hz, 1H, H-6), 4.57–4.44 (m, 2H, H-11′), 3.21 (d, *J* = 4.2, 1.5 Hz, 1H, H-3′), 2.78 (s, 3H, C-2Me), 2.29 (dt, *J* = 14.8, 6.0 Hz, 1H, H-6′), 2.16 (s, 1H, H-5′), 2.08–2.02 (m, 2H, H-4′, H-1′), 1.94 (m, 1H, H-10′), 1.77 (m, 1H, H-10′), 1.67 (d, *J* = 9.8 Hz, 1H, H-6′), 1.32 (s, 3H, H-9′), 0.98 (s, 3H, H-8′). ^13^C-NMR (126 MHz, CDCl_3_) δ 166.0 (C=O), 160.0 (C-3), 149.5 (C-2), 135.9 (C-8), 135.1 (C-8a), 128.3 (C-5), 128.2 (C-6), 127.1 (C-4), 123.6 (C-4a), 121.9 (C-3), 61.9 (C-3′), 62.0 (C-11′), 55.5 (C-2′), 43.7 (C-10′), 40.8 (C-1′), 40.0 (C-5′), 34.0 (C-7′), 27.7 (C-6′), 26.9 (C-4′), 25.9 (C-9′), 25.5 (C-2Me), 20.3 (C-8′). [M + H]^+^: 386.1523 *m*/*z* for calculated C_22_H_25_ClNO_3_, found 386.1525 *m*/*z*.

### 3.17. Plasmodium falciparum Assay

**a**. ***Pf*3D7:** 100 μL of complete media (Sigma-Aldrich’s RPMI 1640, 25 mM HEPES, 10 μg/mL gentamicin, 0.5 mM hypoxanthine supplemented with NaHCO_3_ and 0.5% Albumax II, 37 °C) containing compounds (50–0.3 µM) or 100 µM chloroquine were added to 96 well plates in triplicates. This was followed by the addition of 100 µL erythrocytic asexual culture of *Pf*3D7 (10% hematocrit and 0.25% ring stage parasitemia), maintained at atmospheric conditions of 1% oxygen, 5% carbon dioxide, and 94% nitrogen, and allowed to grow for 96 h at 37 °C. The cultures were then frozen at −80 °C overnight and thawed at 37 °C for 4 h. Aliquots (100 µL) from each well were transferred to black 96-well plates, 100 μL of 2× SYBR Green (Molecular Probes) in lysis buffer (20 mM Tris at pH 7.5, 5 mM EDTA, 0.008% saponin, 0.08% Triton X-100) was added to each well and mixed thoroughly with a pipette. The plates were incubated in the dark at room temperature for 1 h and read at λ_ex_ 485 nm and λ_em_ 530 nm [27]. Infected and untreated red blood cells (iRBC) cultures served as negative control. Percent inhibitory activities were calculated using (=100 − 100*((Sample − 100 µM CQ)/(iRBC − 100 µM CQ)) and used to plot sigmoidal inhibition curves to obtain the EC_50_ values.

**b**. ***Pf*NH54 and *Pf*K1:** The growth inhibitory activities of the compounds against erythrocytic stages of *Pf*NH54 and *Pf*K1 were determined using the [^3^H]-hypoxanthine incorporation assay as previously reported [28]. Parasites in RPMI 1640 medium containing 5% Albumax (without hypoxanthine) were treated with compounds (50–0.3 µM) in 96 well plates for 48 h at 37 °C in 92% N_2_, 5% CO_2_, and 3% O_2_. ^3^H-hypoxanthine (0.5 μCi) was added to each well, incubated for an additional 24 h, harvested onto glass-fiber filters, and washed with distilled water. Radioactivity at each compound concentration was counted using a Betaplate™ liquid scintillation counter (Wallac, Zurich), and the results were recorded as counts per minute (CPM) per well. The CPMs were expressed as a percentage of the CPMs from the untreated controls and used to estimate the EC_50_ from sigmoidal inhibition curves. Artemisinin and CQ were used as positive controls.

### 3.18. Cytotoxicity Assay

Human hepatocarcinoma cell line (Hep G2, ATCC CRL-11997TM) was used for cytotoxicity studies as previously described [29]. The cells were grown in complete medium (DMEM:F12 containing l-glutamine and sodium bicarbonate, 10% FBS, 1% penicillin/streptomycin, Thermo Fisher Scientific) at 37 °C in 5% CO_2_ environment. Cells (198 µL, 5 × 105/mL) were seeded into 96-well plates and incubated overnight. The cells were treated with the compounds (160–1.25 µM, in triplicates) or DMSO (1%) for 72 h. The cell medium was removed and replaced with DMEM:F12 medium containing MTT (0.5 mg/mL) was added to the cells and incubated for 1.5 h. The MTT-containing medium was gently removed and replaced with DMSO (200 µL/well). The contents of each well were then repeatedly mixed with a multichannel pipette and incubated for 10 min. The plates were read at 570 nm. DMSO-treated (1%) cells were used as negative assay control and SDS (10%) was used as assay positive control. Percent cell viabilities were expressed as a percentage of the mean viability of DMSO-treated (1%) cells.

## Data Availability

Data is contained within the article or supplementary material.

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
