# Peer review of "Nopol-Based Quinoline Derivatives as Antiplasmodial Agents"

_molecules, 2021, doi:10.3390/molecules26041008_

Round 1
Reviewer 1 Report
The authors describe the synthesis of nopol-based quinoline derivatives and their antiplasmodial agents. Overall, it is difficult to read the manuscript. The supplementary files are poor prepared and not necessary. The manuscript is not publishable. Here, are some notes. The synthestic routes are not well written and presented. The methods described are wrong in parts. In chapter 3.1. you did not describe the synthesis. You have to give all chemicals with their quality and producer. The solvents for NMR measuremnet are missing. The purification step is poorly described. Do you have experience in culture studies? Why you used SDS as control? Normally you use ethanol or triton-X. Did you treated your cells with your compound in DMSO? This is not right.
Author Response
Reviewer: Overall, it is difficult to read the manuscript.
Authors: Thank you for this observation. We believe that the manuscript is well-written, and each section of the manuscript is annotated correctly. The prose is lucid with proper use of the English Language, and the presentation is appropriate for the study described in the manuscript. Nevertheless, we have made changes to the manuscript, and we hope it is much better.
Reviewer: The supplementary files are poor prepared and not necessary.
Authors: Thank you. We believe that supplementary data is quite important for almost all research articles. We have made appropriate improvements to the supplementary document.
Reviewer: The manuscript is not publishable.
Authors: Thank you.
Reviewer: Here, are some notes. The synthestic routes are not well written and presented.
Authors: Thank you for your observation. We believe the synthetic steps described are appropriate for the study. Please see section 3.2-3.16.
Reviewer: The methods described are wrong in parts. In chapter 3.1. you did not describe the synthesis.
Authors: Thank you for your observation. We have removed the word ‘synthesis’ from section 3.1.
Reviewer: You have to give all chemicals with their quality and producer.
Authors: Thank you for your observation. We have provided the sources of the reagents used. Please see section 3.2-3.15.
Reviewer: The solvents for NMR measuremnet are missing.
Authors: Thank you. NMR solvents are listed as part of the NMR Data and in the supplementary information.
Reviewer: The purification step is poorly described.
Authors: Thank you. The purification methods are properly stated for the methods used and for the type of study.
Reviewer: Do you have experience in culture studies? Why you used SDS as control? Normally you use ethanol or triton-X. Did you treated your cells with your compound in DMSO? This is not right.
Authors: Thank you.
a. Yes, we have significant experience in a variety of biological cultures and assays. SDS is a cytotoxic agent. Please see the following references:
Dong, L., Witkowski, C.M., Craig, M.M., Greenwade, M.M. and Joseph, K.L., 2009. Cytotoxicity effects of different surfactant molecules conjugated to carbon nanotubes on human astrocytoma cells. Nanoscale research letters, 4(12), pp.1517-1523.
Inácio, Â. S., Mesquita, K. A., Baptista, M., Ramalho-Santos, J., Vaz, W. L., & Vieira, O. V. (2011). In vitro surfactant structure-toxicity relationships: implications for surfactant use in sexually transmitted infection prophylaxis and contraception. PloS one, 6(5), e19850. https://doi.org/10.1371/journal.pone.0019850
It is unusual to use ethanol as a cytotoxicity control, but we know that ethanol is toxic to cells at high concentration. SDS and triton-x are surfactants and cytotoxic.
b. Stock solutions of compounds were prepared in DMSO. The final concentration of DMSO in the assay is 1%. The assay description has been modified so that the previously understated points are now more apparent.
Reviewer 2 Report
The manuscript by Ogungbe and co-workers describes nopol-based quinoline derivatives and their inhibitory activity against Plasmodium falciparum. The synthetic methodologies used to prepare compounds 1-16 are not new. Moreover the results obtained in the bioactivity of these compounds were not very promising. Thus, although well written and the compounds well characterized this manuscript lacks novelty. The authors could present further results to explain the activity of some of the new compounds and try to establish some structure activity relationships. Without addressing this issues the manuscript cannot be accept to publish in Molecules.
Author Response
Reviewer: The synthetic methodologies used to prepare compounds 1-16 are not new.
Authors: Thank you for this observation. It is not our intention to develop new synthetic methodologies.
Reviewer: Moreover the results obtained in the bioactivity of these compounds were not very promising.
Authors: Thank you very much. We believe the results have sound scientific premise and merit and it will add to the body of knowledge on antiplasmodial agents.
Reviewer: Thus, although well written and the compounds well characterized this manuscript lacks novelty.
Authors: Thank you very much. We believe that the structures and bioactivity discussed in the manuscript are novel and a welcome extension of knowledge on antiplasmodial agents.
Reviewer: The authors could present further results to explain the activity of some of the new compounds and try to establish some structure activity relationships.
Authors: Thank you very much. We believe that the results and discussion section of the manuscript is a description of structure-activity relationships. Follow-up studies by us and others will undoubtedly provide additional knowledge and context to the current results.
Reviewer: Without addressing this issues the manuscript cannot be accept to publish in Molecules.
Authors: We sincerely appreciate the time spent in reviewing the manuscript.
Reviewer 3 Report
This manuscript entitled “Nopol-Based Quinoline Derivatives as Antiplasmodial Agents” aimed to investigate nopol-based quinoline derivatives for their inhibitory activity against Plasmodium falciparum. The work has some merits, but there are many errors in this manuscript, which need extensive revisions. Some detailed comments are made as below: 1. Please unify the expressions in the text, like line 38, line 46-47, chloroquine and CQ are the same according line 33. 2. Line 31-35, line 40-42, line 56-57, line 102-105, add references. 3. Table 1, like Mefloquine, please standardize the form 4. Please add the specifications and source of TLC (line 137) and silica gel column chromatography (line 138). 5. The data of NMR should be uniformed, like line 174, δ 169.1(C=O), 150.1 (C-5), 148.6(C-2), 142.2(C-2’)……please check the full text. 6. Like DMF, DIEA, HBTU, HOBT and so on in the article, please define their full names and give the purity 7. Line 177, confirm the express of HRMS: [M-H]+, usually expressed as [M-H]-, or [M+H]+. For compound 1 is [M+H]+. 8. The structure of compound should be as standardized as possible, like compound 1 and 2, can choose “ACS Document 1996” in chemdraw. 9. Line 200, “0.85 (s, 3H) 13C NMR (126 MHz, CDCl3)”. 10. About NMR data of every compound, add the reference as possible, and label the atomic number corresponding to each hydrogen signal 11. Line 259 [M-H] 12. Line 312, [M-Na]+, the in fact is [M+Na]+ 13. Line 340, …13C NMR (126 MHz, CDCl3) 13C NMR (126 MHz, CDCl3)…Author Response
Reviewer: The work has some merits, but there are many errors in this manuscript, which need extensive revisions.
Authors: Thank you very much. We have revised specific parts of the manuscripts based on your suggestions.
Reviewer: Some detailed comments are made as below:
- Please unify the expressions in the text, like line 38, line 46-47, chloroquine and CQ are the same according line 33.
Authors: Thank you. We have made the appropriate change. Please see Line 46
- Line 31-35, line 40-42, line 56-57, line 102-105, add references.
Authors: Thank you. A). Reference ‘2’ is appropriate for Line 31-35. Please see Line 36. B). Reference ‘4-6’ is appropriate for Line 40-42. Please see Line 40. C). Line 57-58 has the appropriate reference (15). D). Reference ‘20’ is appropriate for Line 102-105. We have added references 21-23. Please see Line 120.
- Table 1, like Mefloquine, please standardize the form
Authors: Thank you. We think that we have made the appropriate correction. The comment is a bit unclear.
- Please add the specifications and source of TLC (line 137) and silica gel column chromatography (line 138).
Authors: Thank you. It has been corrected. Please see Line 139-141.
- The data of NMR should be uniformed, like line 174, δ 169.1(C=O), 150.1 (C-5), 148.6 (C-2), 142.2(C-2’)……please check the full text.
Authors: Thank you. It has been corrected.
- Like DMF, DIEA, HBTU, HOBT and so on in the article, please define their full names and give the purity.
Authors: Thank you. The full names are now in the manuscript.
- Line 177, confirm the express of HRMS: [M-H]+, usually expressed as [M-H]-, or [M+H]+. For compound 1 is [M+H]+.
Authors: Thank you. The non-standard adduct notation was used, but we have made the appropriate changes.
- The structure of compound should be as standardized as possible, like compound 1 and 2, can choose “ACS Document 1996” in chemdraw.
Authors: Thank you. We have made changes to all relevant structures.
- Line 200, “0.85 (s, 3H) 13C NMR (126 MHz, CDCl3)”.
Authors: Thank you. It has been corrected.
- About NMR data of every compound, add the reference as possible, and label the atomic number corresponding to each hydrogen signal.
Authors: Thank you. The chemical shifts have been assigned for each hydrogen.
- Line 259 [M-H]
Authors: Thank you. It has been corrected.
- Line 312, [M-Na]+, the in fact is [M+Na]+
Authors: Thank you. It has been corrected.
- Line 340, …13C NMR (126 MHz, CDCl3) 13C NMR (126 MHz, CDCl3)
Authors: Thank you. It has been corrected.
Round 2
Reviewer 1 Report
Thank you for the revision. Now the publication has been improved.
Reviewer 2 Report
The revised version presented by the authors did not address the issues pointed in the last review. Although I agree that the work presented is an addition to the body of knowledge on antiplasmodial agents, the work lacks novelty to be published in Molecules. To be accept the authors should present further results to explain the activity of some of the new compounds and try to establish some structure activity relationships.
Reviewer 3 Report
The authors have revised the manuscript to address most of the points raised by the reviewers, and the acceptance of the work is suggested.